# Circulating *KRAS* G12D but not G12V is associated with survival in metastatic pancreatic ductal adenocarcinoma

Jacob E. Till [1,11], Lee McDaniel[2,11], Changgee Chang [3], Qi Long [1], Shannon M. Pfeiffer [4], Jaclyn P. Lyman[4], Lacey J. Padrón[4], Deena M. Maurer[4], Jia Xin Yu[4], Christine N. Spencer[4], Pier Federico Gherardini [4], Diane M. Da Silva [4], Theresa M. LaVallee[4], Charles Abbott[2], Richard O. Chen [2], Sean M. Boyle [2], Neha Bhagwat[1], Samuele Cannas [1], Hersh Sagreiya[1], Wenrui Li[1], Stephanie S. Yee[1], Aseel Abdalla[1], Zhuoyang Wang[1], Melinda Yin[1], Dominique Ballinger[1], Paul Wissel[1], Jennifer Eads[1], Thomas Karasic [1], Charles Schneider[1], Peter O'Dwyer [1], Ursina Teitelbaum[1], Kim A. Reiss[1], Osama E. Rahma[5], George A. Fisher[6], Andrew H. Ko[7], Zev A. Wainberg[8], Robert A. Wolff[9], Eileen M. O'Reilly [10], Mark H. O'Hara [1], Christopher R. Cabanski [4], Robert H. Vonderheide [1] & Erica L. Carpenter [1] ✉

While high circulating tumor DNA (ctDNA) levels are associated with poor survival for multiple cancers, variant-specific differences in the association of ctDNA levels and survival have not been examined. Here we investigate *KRAS* ctDNA (ctKRAS) variant-specific associations with overall and progression-free survival (OS/PFS) in first-line metastatic pancreatic ductal adenocarcinoma (mPDAC) for patients receiving chemoimmunotherapy ("PRINCE", NCT03214250), and an independent cohort receiving standard of care (SOC) chemotherapy. For PRINCE, higher baseline plasma levels are associated with worse OS for ctKRAS G12D (log-rank p = 0.0010) but not G12V (p = 0.7101), even with adjustment for clinical covariates. Early, on-therapy clearance of G12D (p = 0.0002), but not G12V (p = 0.4058), strongly associates with OS for PRINCE. Similar results are obtained for the SOC cohort, and for PFS in both cohorts. These results suggest ctKRAS G12D but not G12V as a promising prognostic biomarker for mPDAC and that G12D clearance could also serve as an early biomarker of response.

Plasma circulating tumor DNA (ctDNA) is increasingly used as a prognostic marker for patients with solid tumors, with low or undetectable pre-therapy levels and on-therapy decreases or clearance associated with improved overall and progression-free survival (OS/ PFS)[1–6]. Such non-invasive biomarkers are urgently needed for metastatic pancreatic adenocarcinoma (mPDAC) where patients do not typically undergo surgical resection and obtaining adequate biopsy tissue for molecular analysis is difficult. Moreover, circulating

[1]Abramson Cancer Center of the University of Pennsylvania, Philadelphia, PA, USA. [2]Personalis, Inc., Menlo Park, CA, USA. [3]Indiana University School of Medicine, Indianapolis, IN, USA. [4]Parker Institute for Cancer Immunotherapy, San Francisco, CA, USA. [5]Dana-Farber Cancer Institute, Boston, MA, USA. [6]Stanford University, Stanford, CA, USA. [7]University of California, San Francisco, San Francisco, CA, USA. [8]University of California, Los Angeles, Los Angeles, CA, USA. [9]The University of Texas MD Anderson Cancer Center, Houston, TX, USA. [10]Memorial Sloan Kettering Cancer Center, New York, NY, USA. [11]These authors contributed equally: Jacob E. Till, Lee McDaniel. ✉e-mail: erical@upenn.edu

biomarkers such as CA19-9 are limited in that 8–15% of PDAC patients produce no detectable CA19-9[7].

PDAC is expected to be the second leading cause of cancer deaths in <10 years[8] and new therapeutic strategies are urgently needed. Current therapeutic options for patients with mPDAC are chemotherapy and select targeted therapies for a small percentage of patients, with few new drug approvals in the last two decades. While immune checkpoint inhibitors (ICI) have dramatically changed the landscape of first-line therapy for patients with advanced non-small cell lung cancer (NSCLC) and other solid tumors[9,10], PDAC tumors have remained largely refractory to immune therapy[11]. One possible exception is the phase 2 PRINCE trial in which a subset of patients receiving nivolumab and gemcitabine/nab-paclitaxel met the primary endpoint of one-year overall survival (OS) compared to a historical 1-year OS[11–13]. As new classes of therapy continue to be evaluated in clinical trials of patients with metastatic PDAC, the prognostic value and clinical utility of non-invasive biomarkers must be rigorously evaluated.

KRAS mutations are present in ~90% of patients with mPDAC, with G12D and G12V comprising the majority of detected KRAS mutations[14]. KRAS variant-specific phenotypes have recently been examined using a pre-clinical model in which CRISPR-based engineering was used to create KRAS-mutant mice with codon 12 and 13 mutations[15]. In human patients, the influence of tissue-based KRAS mutations on survival has been evaluated in multiple studies. Consistent with previous studies[16,17], a multi-institution study of patients with resectable PDAC reported that median OS differed significantly for patients with various KRAS mutations compared to KRAS wild-type[18]. A recent analysis of >600 patients with metastatic PDAC found significantly worse OS when KRAS G12C was detected in a patient's tumor versus KRAS negative. No meaningful survival difference was found for tumors with G12D vs G12V detected[14]. While this study included tissue and liquid biopsy data, only the influence on survival of the presence of a mutation in tissue or plasma was assessed. No analysis of the association of plasma levels of different ctDNA KRAS mutations (ctKRAS) with survival was performed. Other recent studies have focused on the association of baseline and on-therapy levels of plasma ctKRAS with survival for PDAC, but none have examined the associations at a variant-specific level[5,19].

The determinants of driver mutation ctDNA levels have not been clearly elucidated. Plasma ctDNA levels are typically higher for patients with mPDAC than patients with local disease[20]. We and others have shown that disease site may be associated with ctDNA levels. In a study of patients with NSCLC, we demonstrated that patients with extrathoracic disease had significantly higher ctDNA levels than those with intrathoracic disease. Intriguingly, patients with liver metastases had highest ctDNA levels[21]. Hepatic and renal function have been implicated in ctDNA degradation and clearance from peripheral circulation[22]. In advanced melanoma, number of metastases, sum of tumor diameters, and lactate dehydrogenase (LDH) were correlated with ctDNA levels[2]. Mathematical models identified tumor burden and cell death rate as contributors to ctDNA shedding[23], while another group utilized in vitro and in vivo models to identify apoptosis and necrosis as contributors to ctDNA release[24]. In a separate study, the association of tumor volume with ctDNA levels differed in a gene-dependent manner, with tumor volume for KRAS-mutant tumors most strongly associated with ctDNA levels but no significant association for EGFR-mutant tumors[25]. This study did not look at variant-level differences and, to our knowledge, variant-specific associations of ctDNA levels and survival have not been reported for any cancer.

As new therapeutic modalities become available for treating mPDAC, blood-based, prognostic biomarkers obtained prior to or early on therapy will become essential. To date, the assumption has been that lower or undetectable baseline ctDNA or on-therapy ctDNA clearance are almost uniformly associated with better clinical outcomes, regardless of gene or variant. However, data supporting this hypothesis has been limited.

Herein, we investigate variant-specific associations of plasma ctKRAS levels with survival, as well as determinants of ctDNA levels for KRAS driver variants G12D and G12V, among patients with mPDAC undergoing first-line chemoimmunotherapy on the PRINCE clinical trial[2,11]. Our results are replicated using an independent cohort of patients with mPDAC receiving first-line standard of care chemotherapy.

## Results

### Patient characteristics

One hundred twenty-nine patients with first-line mPDAC were enrolled onto the multi-institution PRINCE phase 2 chemoimmunotherapy trial (PICI0002, NCT03214250) (Fig. 1A, Supplementary Table 1)[11,13]. Using a high-sensitivity, pre-amplified ddPCR assay[26], ctKRAS variants were detected for 86 of 115 (74.8%) patients for whom baseline plasma was obtained prior to cycle one day one (C1D1) of therapy. For the 19 PRINCE patients with detectable ctKRAS who received therapy prior to trial enrollment, median ctKRAS variant allele fractions (VAFs) were significantly lower than for 67 patients with de novo stage IV disease (p < 0.0001) (Supplementary Fig. 1); therefore, patients with prior treatment for localized disease were excluded from further analysis. For the 67 patients with a baseline variant detected and no prior treatment, median OS and PFS were 8.7 (95% CI 7.2–10.6) and 6.1 (5.3–7.8) months respectively.

An independent cohort of 85 therapy-naive patients with first-line mPDAC receiving standard of care (SOC) chemotherapy (see Supplementary Table 2 for treatment details) was enrolled at our institution. All patients had baseline plasma obtained prior to C1D1 (Fig. 1B), and ctKRAS detected in 69 samples (81.2%). Median OS and PFS were 8.2 (6.3–12.3) and 3.3 (2.1–5.3) months respectively. PRINCE and SOC patients did not differ significantly in sex, race, ethnicity, or ctKRAS variant detected. There was a significant difference in age (p = 0.0318) and ECOG performance status (p = 0.0011) as the PRINCE trial only enrolled patients with performance status ≤1 and the SOC cohort did include 15 patients with scores >1 (Supplementary Table 2).

Within the PRINCE and SOC cohorts, there was no significant difference in patient characteristics between patients with KRAS G12D- versus G12V-bearing tumors (Supplementary Table 3). There were also no variant-specific differences in RECIST-based tumor measurements (described in Methods) or location of primary tumor within either cohort (Supplementary Figs. 2 and 3). Among clinical laboratory values measured, the liver enzymes ALT and AST were significantly higher for PRINCE patients with ctKRAS G12V versus G12D, although these differences were not seen in the SOC cohort. (Supplementary Figs. 2 and 3). Consistent with a recent, large study of tissue NGS results for mPDAC[14], there was no significant difference in OS or PFS for either PRINCE or SOC patients (or the combined PRINCE and SOC cohort) with a KRAS G12D- versus G12V-bearing tumor (Supplementary Fig. 4). These latter results suggest that, in our cohorts, whether the KRAS driver mutation is G12D vs G12V is not a determinant of survival.

### Prognostic value of baseline ctKRAS levels

We first evaluated the association of baseline plasma ctKRAS VAF levels and survival. Consistent with what we and others have previously shown[19,20], when all 67 evaluable PRINCE patients with any KRAS variant (detectable by our ddPCR assay) were analyzed as a group, ctKRAS VAF levels were significantly associated with OS, with a median 10.2 months for patients with ctKRAS VAF ≤ median and 7.2 months for VAF >median (Cox Regression Hazard Ratio [HR] = 1.85 [95% CI 1.08–3.16], log-rank p = 0.0219). Similar results were achieved for PFS (Supplementary Fig. 5A, D). Next, we repeated the analysis in a variant-specific manner, focusing on ctKRAS G12D and G12V, which were detected for a combined 68% of PRINCE patients (Supplementary Table 2). Given no

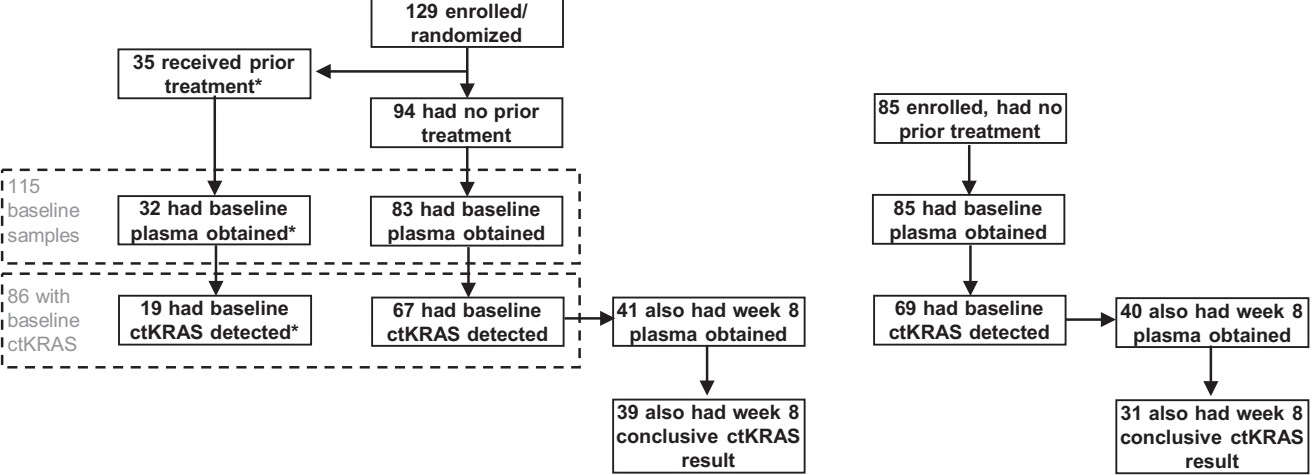

## A PRINCE          B Standard of care (SOC)

* PRINCE patients who received prior treatment were excluded from further analysis due to significantly lower baseline ctDNA-based *KRAS* mutation levels

**Fig. 1 | CONSORT diagram showing the PRINCE clinical trial and standard of care (SOC) validation cohorts.** Plasma samples were obtained from 200 total patients, including 115 PRINCE (**A**) and 85 SOC (**B**) patients. Altogether, 281 samples were analyzed, including: 115 PRINCE samples at baseline (32 for patients with prior therapy plus 83 for patients with no prior therapy), 41 PRINCE samples at week 8 (for patients with no prior therapy), 85 SOC samples at baseline, and 40 SOC samples at week 8. All SOC patients were therapy-naive. Among the 67 PRINCE patients with ctKRAS detected in baseline plasma, 41 also had a week 8 plasma obtained but 26 were unavailable due to the following reasons: 15 discontinued

treatment prior to week 8, 6 died prior to week 8, 3 received a dose at week 8 but there was no blood drawn, and 2 did not receive a dose at week 8 due to an adverse event. Among 69 SOC patients with ctKRAS detected in baseline plasma, 40 also had a week 8 plasma obtained but 29 were unavailable due to the following reasons: 8 either discontinued treatment or were lost to follow-up, 13 died prior to week 8, 5 had a follow-up blood draw outside of the week 8 window, and 3 did not have week 8 samples available at the time of analysis. Source data are provided as a Source Data file.

significant difference in median ctKRAS VAF for G12D vs G12V for PRINCE (p = 0.1520) or SOC (p = 0.1823) or the combined cohorts (p = 0.8728) (Supplementary Fig. 6), we first dichotomized at median VAF. For 33 PRINCE patients with a *KRAS* G12D variant, baseline ctKRAS VAF levels were significantly associated with OS (18.1 vs 6.4 months, HR = 4.18 [1.68–10.39], log-rank p = 0.0010, Fig. 2A) but not for the 23 patients with G12V (8.3 vs. 6.8 months, HR = 1.18 [0.50–2.79], log-rank p = 0.7101, Fig. 2B). Similar results were achieved for PFS (Fig. 2C, D). Importantly, ctKRAS G12D remained significantly associated with OS when analyzed as a continuous variable alone (HR = 1.59 [1.01–2.51], p = 0.046) and in a multivariate analysis (HR = 1.98 [1.10–3.56], p = 0.022), while ctKRAS G12V did not (HR = 0.92 [0.54–1.57], p = 0.765 and HR = 1.10 [0.57–2.11], p = 0.781, respectively, (Supplementary Table 4). ctKRAS G12D VAF also remained significantly associated with PFS, but ctKRAS G12V did not (Supplementary Table 4).

The SOC cohort yielded similar results. For 69 SOC patients with any KRAS mutation detected by our assay (see Methods) in baseline plasma, those with baseline ctKRAS VAF ≤ median had median OS of 12.7 vs 5.5 months (HR = 2.26 [1.36–3.75], log-rank p = 0.0012) (Supplementary Fig. 5B; combined PRINCE and SOC cohort results shown in 5 C). Patients with G12D (12.9 vs 3.2 months, HR = 2.59 1.23–5.44], log-rank p = 0.0096), but not G12V (12.3 vs 7.1 months, HR = 1.48 [0.59–3.69], log-rank p = 0.3981), had a significant association of baseline VAF with OS (Fig. 3A, B). Baseline ctKRAS VAF was significantly associated with PFS overall (Supplementary Fig. 5E; combined PRINCE and SOC cohort results shown in 5 F), and for G12D- but not G12V-bearing tumors for the SOC cohort (Fig. 3C, D; combined PRINCE and SOC cohort results shown in Supplementary Fig. 7). Similar results were obtained using Cox regression analysis for ctKRAS VAF as a continuous variable for univariate and multivariate analysis (Supplementary Table 4). For both cohorts (individually and combined), we tested the possibility of a non-linear relationship between ctKRAS and

survival by plotting the estimated restricted cubic spline function relating to the univariate log ctKRAS Cox models (Supplementary Fig. 8). Additionally, a sensitivity analysis at LODs of 0.10% and 0.25%, levels typically used for clinical NGS ctDNA testing[27] also yielded similar results (Supplementary Table 5). Taken together, these results for both cohorts suggest variant-specific differences in the association of baseline ctKRAS VAF levels with OS and PFS.

**Prognostic value of early, on-therapy changes in ctKRAS levels**
Given that early, on-therapy ctDNA changes have also been associated with outcomes for solid tumors[1,2,5], we next assessed this for PRINCE patients. Among 41 therapy-naive patients with baseline ctKRAS detected and on-therapy plasma, 39 had a conclusive week-8 ctKRAS result ("conclusive results" defined in Methods). However, only 37 patients had any baseline ctKRAS variant detected by our assay, conclusive week-8 ctKRAS result, and sufficient follow-up to determine 1-year OS, the PRINCE trial pre-defined endpoint. Interestingly, all patients experienced a marked decrease in ctKRAS VAF, regardless of whether one-year OS was achieved (Fig. 4A). Therefore, we next analyzed ctKRAS clearance, i.e., ctKRAS detectable at baseline but undetectable at week-8. Among 15 patients with any ctKRAS variant who experienced week-8 ctKRAS clearance, 11 (73.3%) were alive at one-year, whereas, among the 22 patients who did not clear ctKRAS, only 6 (27.3%) were alive, and this difference was significant (p = 0.0084) (Supplementary Fig. 9A). For ctKRAS G12D, the difference was more pronounced, with all patients with ctKRAS G12D clearance alive at one-year but only 12.5% alive when clearance was not achieved (p = 0.0014). For *KRAS* G12V tumors, there was no significant difference in the number of patients alive or dead at one-year by clearance vs no clearance (p = 0.6004) (Fig. 4B). Similar results were obtained for SOC patients (Supplementary Fig. 9B, Fig. 5A, B; combined PRINCE and SOC cohort results shown in Supplementary Fig. 10). This suggests most

**PRINCE Patients**

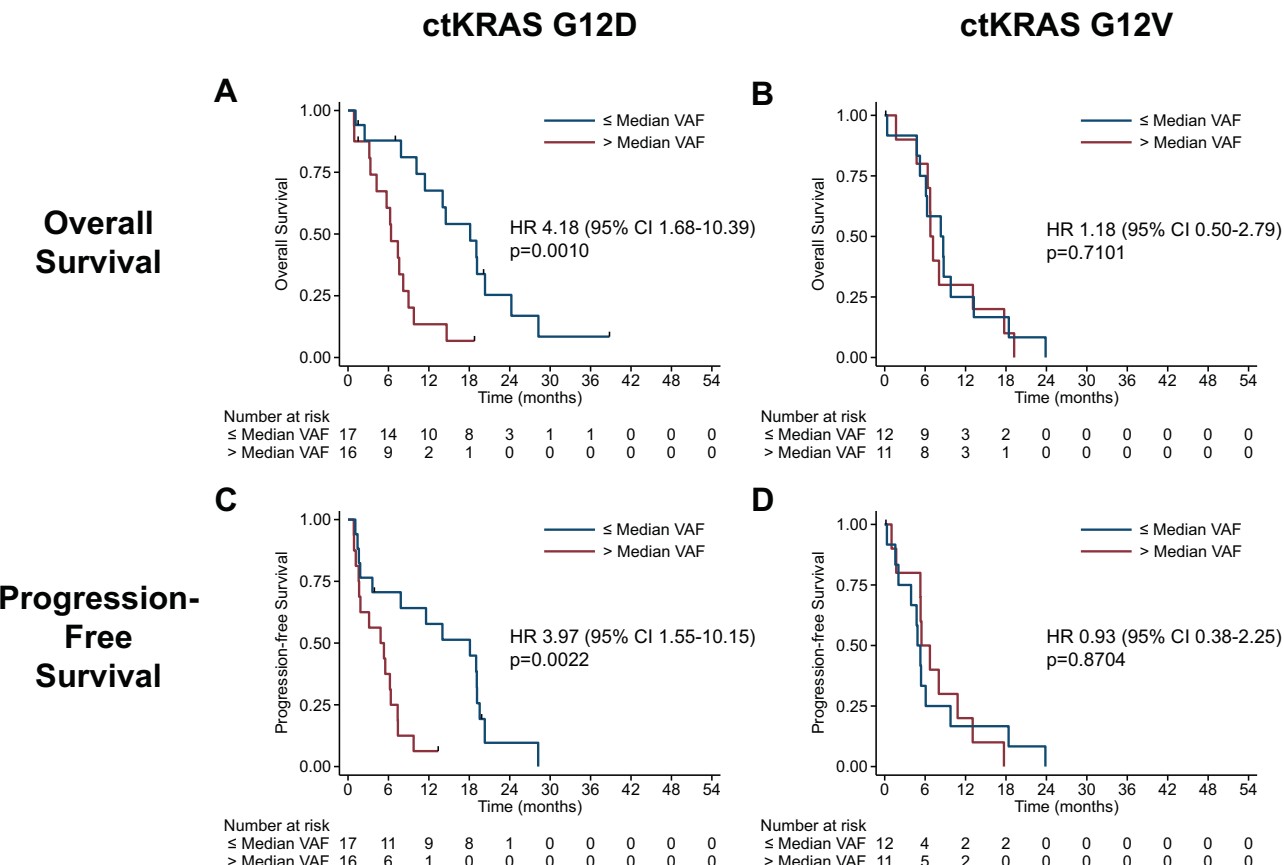

Fig. 2 | **Survival association for baseline ctKRAS variant allele fraction (VAF) by variant for therapy-naive PRINCE patients.** Shown are the Kaplan–Meier curves for baseline VAF dichotomized at the median for overall survival (top, **A**, **B**) and progression-free survival (bottom, **C**, **D**) for patients with G12D- (left, **A**, **C**) or G12V-bearing tumors (right, **B**, **D**). Cox regression hazard ratios (HR) and 95% confidence intervals (CI) are shown with log-rank p-values. Source data are provided as a Source Data file.

patients receiving systemic therapy are likely to experience a decrease in VAF, regardless of one-year survival, whereas ctDNA clearance may be a better prognostic indicator, at least for ctKRAS G12D.

We next assessed whether ctKRAS clearance was associated with OS as a continuous variable for the 39 therapy-naive PRINCE patients with any detectable baseline plasma *KRAS* variant and a conclusive week-8 ctKRAS result. Patients with ctKRAS clearance had a median OS of 18.1 months versus 8.3 months for patients without clearance (HR = 2.81 [1.33–5.92], log-rank p = 0.0048) (Supplementary Fig. 9C). For patients with a *KRAS* G12D variant, ctKRAS clearance was significantly associated with OS (19.0 vs 7.6 months, HR = 12.72 [2.54–63.79], log-rank p = 0.0002, but not for patients with G12V (8.1 vs 8.3 months, HR = 0.62 [0.20–1.92], log-rank p = 0.4058) (Fig. 4C). A similar relationship was found for the entire SOC cohort with a median OS of 16.2 months for patients with ctKRAS clearance and 7.6 months for those without (HR = 3.27 [1.46–7.34], log-rank p = 0.0026) (Supplementary Fig. 9D). In SOC patients with a *KRAS* G12D variant, ctKRAS clearance was associated with OS (16.3 vs 8.2 months, HR = 18.50 [2.21–155.20], log-rank p = 0.0006), but not for those with G12V (8.5 vs 9.17 months, HR = 1.37 [0.33–5.80], log-rank p = 0.6645) (Fig. 5C). ctKRAS clearance was associated with PFS for both cohorts overall (Supplementary Fig. 9E, F), but only for G12D- and not G12V-bearing tumors (Figs. 4D and 5D). Analysis of ctKRAS clearance with respect to OS and PFS for the combined PRINCE and SOC cohorts is shown in Supplementary Fig. 10, 11. These results suggest variant-specific differences in the association of ctKRAS clearance and survival.

## Association of patient clinical characteristics with baseline ctKRAS G12D vs G12V VAF levels

In an exploratory analysis to generate hypotheses as to the biology underlying the variant-specific differences reported above, we next performed a correlation analysis of baseline ctKRAS VAF levels with several baseline clinical variables for PRINCE patients (Fig. 6 and Supplementary Table 6). Among imaging-based tumor measurements, four variables were significantly associated for ctKRAS G12D VAF levels but not G12V ("Sum of diameters–all lesions" (G12D $\rho = 0.4160$, p = 0.0160; G12V $\rho = 0.0633$, p = 0.7743), "Sum of diameters–metastases" (G12D $\rho = 0.3830$, p = 0.0278; G12V $\rho = 0.3386$, p = 0.1140), "Count of lesions–all lesions" (G12D $\rho = 0.4878$, p = 0.0040; G12V $\rho = 0.3459$, p = 0.1060), and "Count of lesions–metastases" (G12D $\rho = 0.5011$, p = 0.0030; G12V $\rho = 0.3459$, p = 0.1060)). Similar associations were seen for these imaging variables in the SOC cohort. Among clinical laboratory values for PRINCE patients, only AST was significantly associated for both G12D and G12V. ALT was significantly associated with baseline ctKRAS VAF for G12V but not G12D. These associations for AST and ALT were not observed in the SOC cohort. SOC patient G12D but not G12V VAF was significantly associated with CA19-9 and albumin, although these associations were not seen for PRINCE patients. No difference in ctKRAS VAF was seen for sex or location of primary tumor (Supplementary Fig. 12). Taken together, these exploratory results suggest possible differences in the association of tumor measurements and clinical laboratory values with ctKRAS VAF levels for subjects with G12D- vs G12V-bearing tumors.

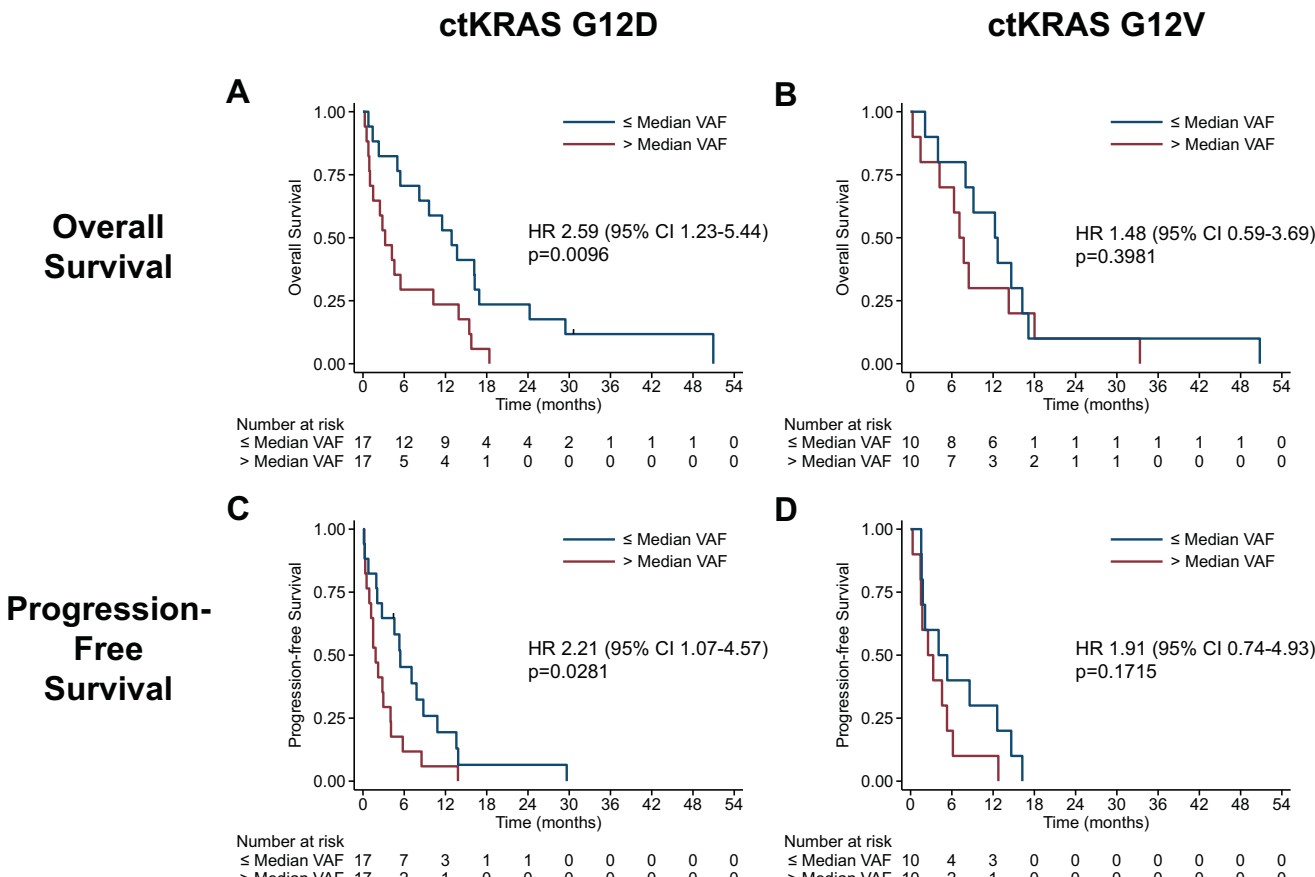

**Fig. 3 | Survival association for baseline ctKRAS variant allele fraction (VAF) by variant for therapy-naive standard of care (SOC) patients.** Shown are the Kaplan–Meier curves for baseline VAF dichotomized at the median for overall survival (top, **A**, **B**) and progression-free survival (bottom, **C**, **D**) for patients with G12D- (left, **A**, **C**) or G12V-bearing tumors (right, **B**, **D**). Cox regression hazard ratios (HR) and 95% confidence intervals (CI) are shown with log-rank p-values. Source data are provided as a Source Data file.

## Discussion

Here we demonstrated that survival outcomes are associated with baseline and on-therapy measures of *KRAS* variants in plasma ctDNA in a variant-specific manner for patients with de novo mPDAC, even after adjusting for clinical covariates. We demonstrated this for a patient cohort enrolled on a multi-institution clinical trial of chemoimmunotherapy (PRINCE), and corroborate this finding in a second, independent patient cohort, also with first-line mPDAC, receiving standard of care therapy. Altogether, our combined cohort comprises 214 patients and 281 total plasma samples analyzed. While others have demonstrated that high baseline or on-therapy ctKRAS levels are associated with worse outcomes for patients with mPDAC[19], we establish a variant-specific association between plasma *KRAS* ctDNA levels and survival. Moreover, while ctDNA levels have been previously evaluated as a prognostic biomarker in the setting of patients with advanced NSCLC receiving ICI therapy[1], we establish a role for ctKRAS as a non-invasive biomarker for patients with mPDAC receiving chemoimmunotherapy. Thus, our findings build on those reported by Padron et al. for the PRINCE trial in which circulating predictive biomarkers, based mainly on T cell subsets, were significantly associated with survival[11].

Studies of lung, melanoma, and other solid tumors have confirmed a strong association of baseline ctDNA levels with OS and tumor volume[2,28,29], suggesting that baseline ctDNA levels might broadly serve as a prognostic biomarker. However, variant-specific associations were not examined in these studies, and if baseline ctDNA levels are increasingly considered as part of a diagnostic work-up for patients with solid tumors, identifying and accounting for any variant-specific differences in association with survival will be essential for clinical implementation and represent a possible set of new stratification factors in clinical trials. Our results clearly demonstrate for the PRINCE and the SOC cohorts that while baseline ctDNA VAF levels for G12D, the most commonly detected *KRAS* variant in PDAC, are strongly associated with OS and PFS, there is no significant OS or PFS association for ctKRAS G12V levels. This initial analysis was performed using the median for each group since ctKRAS VAF medians were not significantly different for patients with *KRAS* G12D- vs G12V-bearing tumors in either cohort. Nevertheless, we performed additional analysis of ctKRAS VAF as a continuous variable, yielding the same results, even after adjusting for clinical covariates. Given that the level of detection (LOD) for our pre-amplified ddPCR assay is quite low at 0.04% (see Methods), we performed additional sensitivity analysis at higher LODs, commensurate with those typically used for ctDNA NGS results, and these results confirmed our findings.

Our results contribute additional, strong evidence for the clinical utility of on-therapy blood draws for patient monitoring. Although variant allele fraction is the most common ctDNA metric shown in physician reports, we demonstrate that almost all patients who received either chemoimmunotherapy or chemotherapy experienced large decreases early on therapy (week 8), regardless of whether the clinical endpoint of one-year survival was achieved. This suggests that the increasingly common practice of comparing serial VAF percent

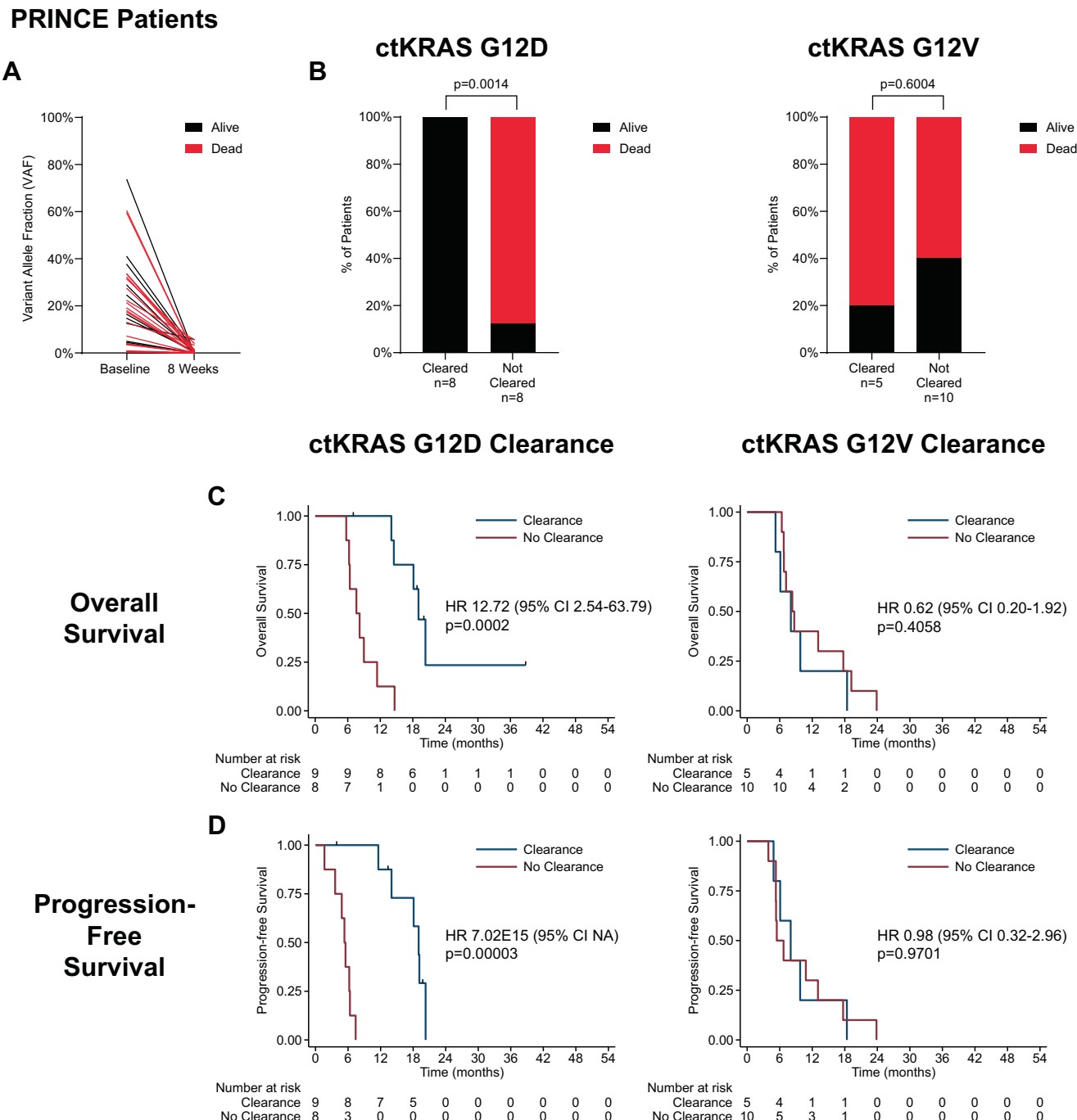

**Fig. 4 | Survival association with early, on-therapy ctKRAS dynamics by variant for PRINCE cohort.** Shown is the association with one-year survival (alive or dead at one year) for therapy-naive PRINCE patients with any detected baseline ctKRAS mutation as measured by **A** changes in ctKRAS variant allele fraction (VAF) from baseline to week 8 on therapy (*n* = 37), or **B** ctKRAS clearance at week 8 on therapy for G12D only (left) and G12V only (right). Shown in **C** is Kaplan-Meyer analysis dichotomized by ctKRAS clearance vs no clearance and association with overall survival for G12D only (left), and G12V only (right). Results for progression-free survival shown in **D**. Among the 39 PRINCE patients with both a baseline and week 8 plasma obtained, 2 had insufficient follow-up to determine 1-year survival and were thus excluded from the results shown in **A**, **B**. Mann–Whitney test (two-sided) used for comparisons in **A**. Fishers exact test (two-sided) used in **B**. Cox regression hazard ratios (HR) and 95% confidence intervals (CI) are shown with log-rank p-values for **C**, **D**. Source data are provided as a Source Data file.

change for a patient over the course of therapy, including in the physician reports from commercial ctDNA laboratories, is not always an accurate means of predicting response. Instead, our results suggest that ctDNA clearance is a more accurate on-therapy biomarker. In addition, and similar to what was described above for baseline ctDNA, our results suggest that consideration must be given to the specific variant being tracked for ctDNA-based disease monitoring. For the PRINCE and SOC cohorts studied here, variant-specific differences in

the association of ctDNA clearance and OS and PFS were strongly evident. Interestingly, the effect size for ctKRAS clearance was larger than for baseline ctKRAS. For example, for PRINCE patients we observed a hazard ratio of 4.18 [95% CI 1.68–10.39] for the association of ctKRAS G12D VAF with overall survival, whereas the hazard ratio for this cohort for ctKRAS clearance and overall survival was 12.72 [95% CI 2.54–63.79]. Similar differences were seen for the SOC cohort and for PFS for both. Of note, prospective, ctDNA-based adaptive trials already

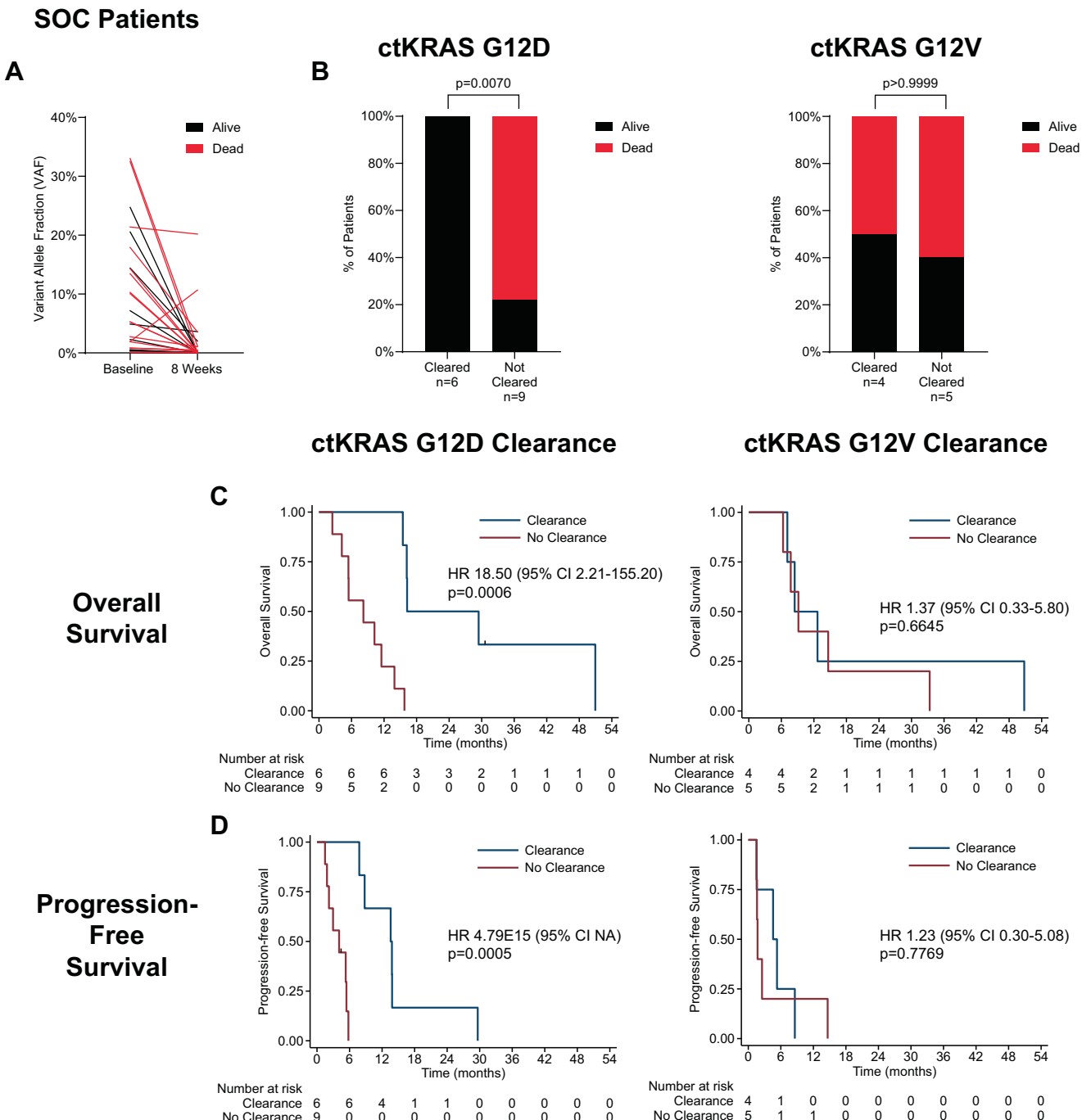

**Fig. 5 | Survival association with early, on-therapy ctKRAS dynamics by variant for SOC cohort.** Shown is the association with 1-year survival (alive or dead at one year) for therapy-naive SOC patients with any detected baseline ctKRAS mutation as measured by **A** changes in ctKRAS variant allele fraction (VAF) from baseline to week 8 on therapy (*n* = 31), or **B** ctKRAS clearance at week 8 on therapy for G12D only (left) and G12V only (right,). Shown in **C** is Kaplan–Meyer analysis dichotomized by ctKRAS clearance vs no clearance and association with overall survival for G12D only (left), and G12V only (right). Results for progression-free survival shown in **D**. Mann–Whitney test (two-sided) used for comparisons in **A**. Fishers exact test (two-sided) used in **B**. Cox regression hazard ratios (HR) and 95% confidence intervals (CI) are shown with log-rank *p* values for **C**, **D**. Source data are provided as a Source Data file.

underway (NCT05281406, NCT04093167) will be essential for determining the true clinical benefit of on-therapy ctDNA monitoring. For example, if a patient has not cleared ctDNA by week 8, this may indicate a need to escalate therapy if tolerable or change therapy. Design of such trials must be informed by studies such as ours elucidating variant-specific differences in the association of ctDNA clearance with OS and PFS. Taken together, this work further establishes a role for on therapy ctDNA monitoring for a cancer such as mPDAC for which imaging can be an imperfect approach.

Additional studies, perhaps utilizing recently developed pre-clinical models[15], will be necessary to identify the biological under-pinnings of our somewhat perplexing finding of variant-specific differences in the association of ctKRAS levels and survival. Subjects with mPDAC with KRAS G12D-bearing tumors had no difference in survival compared to those with G12V in either of our cohorts, or in a recently published large study[30]. Moreover, ctKRAS VAF levels were not significantly different within each of our cohorts when compared for ctKRAS G12D vs G12V. Our exploratory analysis of RECIST tumor

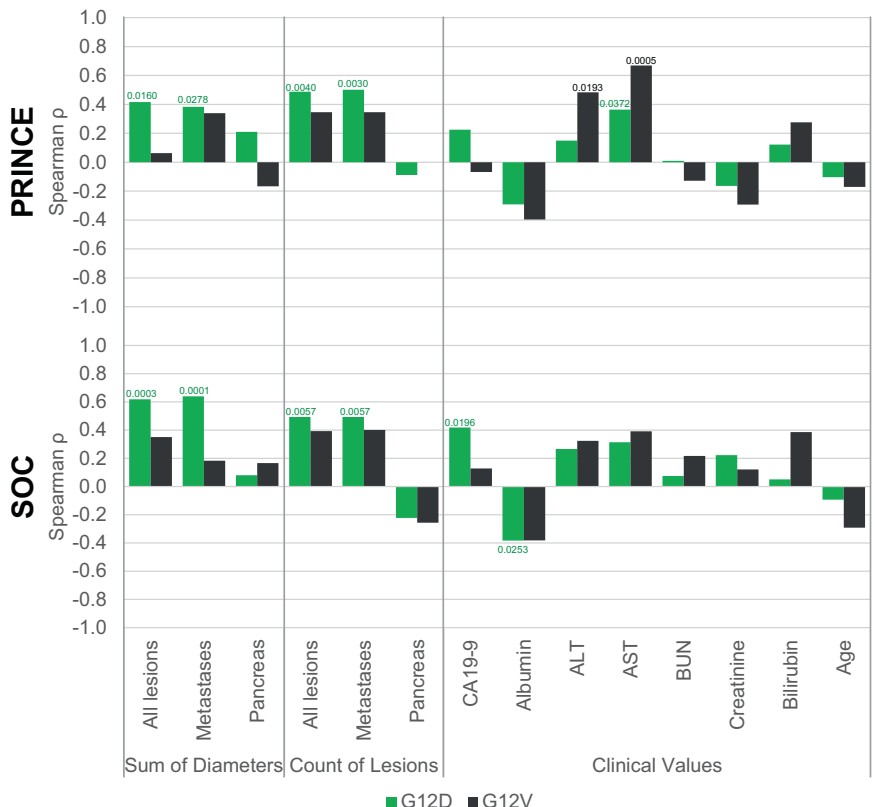

**Fig. 6 | Correlation of clinical variables with baseline ctKRAS variant allele fraction (VAF) levels for therapy-naive PRINCE and standard of care (SOC) patients.** Shown are results for Spearman correlations (two-sided) for all continuous clinical variables with ctKRAS VAF levels for PRINCE patients (top) and SOC patients (bottom). For each variable, the Spearman ρ is shown for patients with ctKRAS G12D (green bars) and G12V (dark gray bars). "Sum of Diameters" and "Count of Lesions" refer to categories of RECIST imaging measurements (see Methods). Among 56 total PRINCE patients analyzed, 33 have ctKRAS G12D and 23 have G12V. For SOC, the number of patients varies slightly by clinical variable (see Supplementary Table 5 for exact patient numbers). Significant p-values are indicated for the bar over which it appears. If only one bar in a pair has a p-value, that indicates significance for the association of the clinical variable with that particular ctKRAS variant (G12D or G12V) VAF levels but not for the other variant. If a p-value appears over both bars in a pair, that indicates that the association of that clinical variant with ctKRAS VAF for both G12D and G12V was significant. ALT alanine transaminase, AST aspartate transaminase, and BUN blood urea nitrogen are used. Source data are provided as a Source Data file.

measurements and clinical laboratory values did reveal intriguing hypothesis-generating findings, although the extendibility of these results would have to be confirmed through analysis of larger data sets for which multivariate linear regression or other more sophisticated analysis would be more suitably powered. Given that previous studies have demonstrated an association between tumor volume and ctDNA levels[2,23,28], the positive and significant association of ctKRAS G12D levels with four of the six baseline tumor volume measurements for both patient cohorts was not unexpected. However, the lack of association between ctKRAS G12V levels and any of the tumor volume measurements was unanticipated and should be further studied. Intriguingly, measures of pancreatic lesions were not significantly associated with ctKRAS levels for either variant in either cohort. Although this suggests the possibility that measures of metastatic, rather than primary, tumor may be more strongly associated with ctKRAS levels overall, our analysis was not designed to draw comparisons between clinical measurements. Consistent with previous studies showing an association between liver function and ctDNA levels[22], AST levels were significantly associated with both G12D and G12V ctKRAS levels for PRINCE patients. However, there was no association of AST with baseline VAF levels for ctKRAS G12D or G12V for the SOC cohort. These provocative but exploratory results are limited by sample size yet provide initial hints to underlying differences in the drivers of ctDNA levels overall and by ctKRAS variant. These results must be further studied with larger patient cohorts, multi-omic translational datasets, and together with pre-clinical models in which additional

measures of cell turnover, such as mitotic rate and cell death mechanism, can also be measured.

Strengths of our study include the multi-institution enrollment of PRINCE trial participants and central processing of all plasma samples. We validated our results showing a variant-specific association between baseline ctKRAS and ctKRAS clearance with survival for the PRINCE cohort in an independent cohort of patients receiving first-line standard of care therapy. Moreover, even when the two cohorts were combined, ctKRAS G12D, but not G12V, remained an independent predictor of survival. This research applied a widely utilized, well-studied, and commercially available ddPCR approach to ctKRAS variant detection which can easily be applied and, thus, replicated and validated in other institutions. ddPCR is inexpensive and has a rapid turn-around time compared to NGS analysis of ctDNA. Nevertheless, our study also had limitations. Matched PBMCs were not available to rule out clonal hematopoiesis (CHIP) as a source of the plasma ctKRAS variants detected for our patients[31]. Regardless, CHIP is unlikely to be a confounder of our results as concordance with matched tissue for the PRINCE cohort was quite high at 85.5% (see Methods), and all patients with a ddPCR result at baseline and week 8 experienced a decrease in ctKRAS VAF, many of them substantial. Each of the 3 PRINCE trial arms had insufficient N to determine whether our finding of a variant-specific association between ctKRAS and survival differed significantly from one treatment arm to the other. Given the smaller size of the G12D- and G12V-only cohorts analyzed, some differences in clinical variables may be due to power limitations and sampling error. While

participants in the PRINCE and SOC cohorts were well-balanced for age and gender, lack of racial diversity limits the application of these results to minority populations without further study. Our results bear further validation in an independent, more diverse, and, perhaps, larger prospectively enrolled cohort of mPDAC patients. Validation of the variant-specific prognostic value of this commercially available ddPCR assay using an orthogonal assay, such as widely used plasma targeted NGS testing, will be essential. Further analysis of ctKRAS as a biomarker for patients with resectable or locally advanced PDAC will also be necessary to determine whether survival for those patient cohorts is associated with ctKRAS in a variant-dependent manner. Our study was not sufficiently powered to measure the association of ctKRAS G12C or other *KRAS* variant levels and survival, an analysis that will be essential as *KRAS* targeted therapies are eventually prescribed in the setting of lung, pancreatic, and other tumors[32,33]. Additional studies will extend this analysis to determine whether ctKRAS variant-specific associations exist in other solid tumors such as lung and colorectal cancer where *KRAS* mutations are frequently detected[14].

In conclusion, our multi-institution study demonstrates a variant-specific association of survival with ctKRAS and strengthens the case for use of ctKRAS G12D as a biomarker for patients with mPDAC.

## Methods
### Patients and Samples
As previously described[11], the phase 1b/2 PRINCE study (PICI0002, NCT03214250) was approved by lead (University of Pennsylvania) institutional review board and accepted at all participating sites. It was conducted in compliance with the Declaration of Helsinki and International Conference on Harmonisation Good Clinical Practice guidelines. Written informed consent was provided by all 129 patients with mPDAC prior to enrollment (Supplementary Table 1). Patients were randomized to one of three experimental treatment regimens: nivolumab + chemotherapy, sotigalimab + chemo, or sotigalimab + nivolumab + chemo. Blood specimen collection and processing have been previously described (detailed methods are provided in the Supplementary Information File)[11]. In brief, blood specimens were collected in Streck cfDNA blood collection tubes at baseline and week 8 (prior to C1 and C3 of therapy), processed to plasma and banked at a central facility (Infinity Biologix, Piscataway, NJ, USA) and shipped to the University of Pennsylvania for further analysis. Baseline or archival tumor specimens, survival outcomes, Response Evaluation Criteria In Solid Tumors (RECIST) version 1.1 data, and standard laboratory values were collected from participating sites. For RECIST analysis, target lesions were included in both the sum of diameters and count of lesions measurements. Non-target lesions were not measured but were included in the count of lesions.

Patients in the standard of care (SOC) cohort were enrolled at the Hospital of the University of Pennsylvania (Philadelphia, PA) under IRB Protocol #822028 after obtaining written informed consent. Patients were not compensated for their participation in this biobanking study. All patients had previously untreated metastatic disease and received the chemotherapy described in Supplementary Table 2. Self-reported sex, ethnicity, and race data are presented as collected as part of the PRINCE clinical trial data at participating sites and as recorded in the medical record for SOC patients (individual patient level data is provided in the data supplement). The sex data were considered in determining if there was a difference between the two cohorts (Supplementary Table 2) and considered in the univariate and multivariate Cox analyses (Supplementary Tables 4 and 5). Gender data was not available at the time of data collection and thus not included. The study was conducted in accordance with the Declaration of Helsinki. Blood specimens were collected in K2EDTA vacutainers and processed to plasma within 3 hours or Streck cfDNA tubes and processed within 7 days and banked as previously described[26]. Survival outcomes,

laboratory, and imaging data were abstracted from the electronic medical record. Disease progression events, as described in the medical record by the treating physician (based on radiographic or other clinical evidence) were used for the SOC cohort. Tissue specimens were not available for the SOC cohort. RECIST analysis for the SOC cohort was performed by an experienced radiologist (HS). For the PRINCE cohort, eight patients had no target pancreas lesions identified and, among the eight, three also did not have any non-target lesions identified. These appear as zero values in Supplementary Fig. 2. For the SOC cohort, one patient had neither target nor non-target pancreas lesions identified. These appear as zero values in Supplementary Fig. 3. REMARK reporting guidelines were followed for both cohorts.

Median follow-up for OS for the 83 PRINCE trial and 85 SOC patients with baseline plasma were 9.9 (IQR 6.1–19.0) and 9.4 (IQR 4.2–16.3) months, respectively. Median follow-up for PFS for the 82 PRINCE trial and 85 SOC patients with baseline plasma were 6.7 (IQR 3.6–14.1) and 4.1 (IQR 1.7–10.2) months, respectively. One PRINCE patient with OS data did not have PFS data.

### Circulating cell-free DNA (ccfDNA) extraction, quantification, preamplification and droplet digital PCR (ddPCR)
Extraction of DNA from plasma was performed using the QIAamp Circulating Nucleic Acid Kit (Qiagen, # 55114)[26] or QIAamp MinElute ccfDNA Midi Kit (Qiagen, # 55284) as previously described[34]. Quantification of extracted ccfDNA was performed by qPCR for a 115 bp amplicon of the ALU repeat element[35]. Pre-amplification of DNA was carried out for the *KRAS* G12 locus[26]. For the SOC cohort, ddPCR was performed on the RainDrop (RainDance Technologies, Inc.) platform for 60 of 85 patients as previously described (detailed methods are provided in the Supplementary Information File)[26]. For the PRINCE cohort and the remaining 25 SOC patients, the QX200 (Bio-Rad Laboratories, Inc) platform was utilized. All samples were screened for *KRAS* variants using the ddPCR™ *KRAS* G12/G13 Screening Kit (Bio-Rad, #1863506). Samples were called positive if the variant copy number was greater than 3 standard deviations above the mean for a panel of 20 healthy control samples. Negative samples fell into two categories, conclusive negatives, where DNA concentration and total copies assayed were sufficient to reach a limit of detection of 0.04% variant allele fraction (VAF = mutant copies divided by the sum of mutant and wild-type copies), or inconclusive, when the sample did not meet these criteria. For each patient with at least one positive sample, the samples with the highest variant allele fraction were assayed on 7 individual variant assays (G12A/C/D/R/S/V and 13D) to determine the variant. Each sample was run again on the appropriate single variant assay as these individual assays had lower average false positive rates resulting in greater sensitivity for variant detection. Four PRINCE patients had both a G12D and a G12V *KRAS* variant detected in plasma and these patients were excluded from analysis in which variant-specific differences in OS or PFS were calculated (detailed methods are provided in the Supplementary Information File). Among the 115 PRINCE patients for whom baseline plasma was obtained prior to C1D1 (Fig. 1A), 62 patients had matched tissue DNA or RNA analyzed, resulting in concordance of 53/62 (85.5%) for detected *KRAS* variants. There was no difference in plasma/tissue concordance (Fisher's exact test p > 0.9999) for KRAS G12D (89.3% or 25/28) compared to G12V concordance (87.5% or 21/24). For the 3 discordant G12D and the 3 discordant G12V calls, all were found by the plasma ddPCR assay and not by NGS of matched tissue (Supplementary Table 1).

### Statistics and reproducibility
Kaplan-Meier, log-rank, and Cox regression analyses were performed in Stata/IC 16.1 (Stata Corp.). Spearman correlation, Dunn's multiple comparisons, Fisher's exact, Chi-square and two-tailed Mann-Whitney analyses were performed in Prism 9.5.1 (GraphPad Software). Plots of the estimated restricted cubic spline function relating single

predictors to survival were generated in R (version 4.2.1) using the survival, rms, and ggplots2 packages and combining various functions to, e.g., compute restricted cubic splines, and fit a Cox proportional hazards regression model. Mann-Whitney test was used for comparison of continuous and ordinal variables between two groups. Fisher's exact test was used for comparison of binary variables between two groups. Spearman correlation statistics were used to compare two continuous variables. Dunn's multiple comparisons test was used for comparison of continuous and ordinal variables between more than two groups. Chi-square test was used for comparison of categorical variables with greater than two categories between groups. For survival analyses at baseline, cohorts were dichotomized at less than or equal to the median vs greater than the median ctDNA-based *KRAS* (ctKRAS) VAF. For survival analysis 8 weeks into treatment, cohorts were dichotomized by ctKRAS clearance or not, defined as going from detectable ctKRAS to conclusive negative vs remaining detectable, respectively. Survival statistics were generated using Kaplan-Meier curves to estimate survival function, Log-rank test for comparing survival functions between two groups, and Cox model to estimate hazard ratio between two groups. The association of baseline ctKRAS VAF and survival was not analyzed for each of the three PRINCE trial arms as the patient numbers were insufficient. For the PRINCE cohort, additional univariate and multivariate Cox regression models were fit to log ctKRAS VAF as a continuous variable and covariates (age, sex, ECOG performance status, and sum of diameters of all RECIST target lesions at baseline). Given our cohort sizes, the multivariate analysis was conducted with a limited number of covariates. CA19-9 was not considered for the PRINCE cohort as it was not available for a large proportion of patients at baseline. Similar univariate and multivariate Cox regressions were fit to the SOC cohort including CA19-9, as it was available for >90% of this cohort, but excluding sum of diameters for all RECIST target lesions as this was unavailable for a large proportion of patients in this cohort. ddPCR VAF assay was performed twice for all samples as described above (first with a multi-variant assay, and again with a variant specific assay), variant specific VAF was used for patients with a single variant detected, multi-variant assay results were used if multiple variants were detected.

Tumor burden features assessed for association with ctKRAS VAF levels included sum-of-diameters for RECIST target lesion measurements for pancreas lesions, all metastatic (non-pancreatic) lesions, and all lesions. Additionally, count of all (target and non-target) RECIST lesions for pancreas, all metastatic (non-pancreatic) sites, or all sites were included. The other clinical features assessed for association with ctKRAS levels included albumin, ALT, AST, BUN, creatinine, total bilirubin, age, sex, and primary tumor location (head vs body vs tail).

Whole exome sequencing was performed in the same manner as prior investigations[36]. Capture libraries for whole-exome sequencing were constructed from tumor and blood DNA respectively, with enhanced coverage of notable driver genes and clinically relevant genes. Illumina sequencers (HiSeq 2500 or NovaSeq) were utilized for paired-end sequencing. Reads were aligned to human genome build hs37d5, with ctDNA corollary *KRAS* somatic mutations identified from known loci (G12, G13, Q61) with at least 10 reads, minor allele frequency (MAF) > 2.5%, phred score ≥30. Transcriptome sequencing using Illumina sequencers (HiSeq 2500 or NovaSeq) was aligned to human genome build hs37d5 using STAR and normalized expression values in transcripts per million (TPM) were calculated. In addition, somatic mutations identified from known loci (G12, G13, Q61) were identified using the same parameters as whole-exome sequencing (at least 10 reads, minor allele frequency (MAF) > 2.5%, phred score ≥ 30).

### Reporting summary

Further information on research design is available in the Nature Portfolio Reporting Summary linked to this article.

## Data availability

The individual patient age data are protected and are not available due to data privacy laws. All remaining de-identified data generated in this study are provided in the Source Data file. Source data are provided with this paper.

## Code availability

This study does not use custom code or mathematical algorithms central to the conclusions, therefore no code has been made available.

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

## Acknowledgements

The PRINCE study was sponsored by the Parker Institute for Cancer Immunotherapy (PICI) and funded by the Cancer Research Institute, Bristol Myers Squibb and PICI. We acknowledge research funding from PICI (A.A., A.H.K., D.B., E.L.C., E.M.O., J.E.T., M.H.O., M.Y., N.B., S.C., S.S.Y., Z.A.W., and Z.W.), the University of Pennsylvania Pancreatic Cancer Research Center (Penn PCRC; A.A., D.B., E.L.C., J.E.T., M.Y., N.B., S.C., S.S.Y., and Z.W.), and Nation Institutes of Heath Nation Cancer Institute Abramson Cancer Center of the University of Pennsylvania Core Support Grant (P30CA016520; A.A., D.B., E.L.C., J.E.T., M.Y., N.B., Q.L., S.C., S.S.Y., and Z.W.). We gratefully acknowledge generous support from the James and Marlene Scully Liquid Biopsy Innovation Fund (D.B., E.L.C., J.E.T., M.Y., and S.C.). We thank Drs. Zheng-Lin, Hoyek, and Bekaii-Saab, and their team at Mayo Arizona for helpful discussion.

## Author contributions

J.E.T., L.M., M.Y., and E.L.C. conceived of and supervised experiments. J.E.T., L.M., C.C., Q.L., S.M.P., L.J.P., D.M.M., J.X.Y., C.N.S., P.F.G., C.A., S.M.B., S.C., W.L., C.R.C., and E.L.C. performed data analysis. J.P.L., D.M.D., T.M.L., R.O.C., O.E.R., G.A.F., A.H.K., Z.A.W., R.A.W., E.M.O., M.H.O., C.R.C., and R.H.V. supervised PRINCE clinical trial and clinical data collection. N.B., H.S., A.A., Z.W., and D.B. performed experiments or collected data. J.E.T., H.S., S.S.Y., M.Y., P.W., J.E., T.K., C.S., P.O., U.T., K.A.R., M.H.O., and E.L.C. supervised standard-of-care banking and clinical data collection. J.E.T., C.R.C., and E.L.C. wrote the manuscript. All authors edited the manuscript.

## Competing interests

EMO reports research funding from Genentech/Roche, BioNTech, AstraZeneca, Arcus, Elicio, Parker Institute, NIH/NCI, Digestive Care, Break Through Cancer and consulting/DSMB for Arcus, Ability Pharma, Alligator, Agenus, Boehringer Ingelheim, BioNTech, Ipsen, Merck, Moma Therapeutics, Novartis, Syros, Leap Therapeutics, Astellas, BMS, Fibrogen, Revolution Medicine, Merus, AstraZeneca, BioSapien, Astellas, Thetis, Autem, Novocure, Neogene, Tempus, Fibrogen, Merus, Agios (spouse), Genentech-Roche (spouse), Eisai (spouse). RHV reports he has received consulting fees from BMS, is an inventor on patents relating to cancer cellular immunotherapy, cancer vaccines, and KRAS immune epitopes (10286066, 9555105, 8722400, 7851591, 7754482, and 7385023), and receives royalties from Children's Hospital Boston for a licensed research-only monoclonal antibody. MHO report grants from the Parker Institute for Cancer Immunotherapy (PICI) during the conduct of this study as well as grants from BMS, Celldex, Psioxus, Genmab, Geistlich, Arcus, Elicio; grants and non-financial support from Stand Up To Cancer; and personal fees from Natera, Merus, and Alligator outside the submitted work. OER reports personal fees from Merck, Celgene, Five Prime Therapeutics, GlaxoSmithKline, Bayer, Roche/Genentech, Puretech, Imvax and Sobi, as well as employment at and stock ownership of AstraZeneca outside the submitted work and has a patent pending (DFCI 2386.010) for methods that make use of pembrolizumab and trebananib. PFG reports stock ownership in Teiko.bio. TML reports Coherus Biosciences employment; stock ownership in AstraZenca and Coherus Biosciences. ZAW reports grants from PICI during the conduct of this study as well as research funding from BMS, Arcus, and Plexxikon and outside the submitted work and consulting/DSMB from BMS, Pfizer, Merck, Eli Lilly, Daiichi, AstraZeneca, Arcus, Novartis, Ipsen, Seagen, Alligator, Boehringer Ingelheim, Astellas, Eisai and Genentech/Roche. GAF reports personal fees from Merck, Roche/Genentech and CytomX outside the submitted work; and his spouse owns stock in Seattle Genetics. JXY reports stock in and employment with Bristol Myers Squibb. CWA reports Personalis, Inc. employment. JET reports research funding from PICI during the conduct of this study. KAR reports research funding from Clovis Oncology, BMS, GSK, and Lilly Oncology, membership on advisory boards for Carisma Therapeutics (2021), Astrazenca (2022), BMS (2022 and 2024), Synovation (2024), Merus (2024), Guardant (2024), the Medical Affairs Advisory Council for FMI (2023–2025), and consultanting for MOMA (2024). LDM is a current employee of BillionToOne and former employee of Personalis, a company that PICI paid to produce sequence information for some

samples reported in this paper as part of a collaboration. He reports stock ownership in Personalis. SSY reports employment with Century Therapeutics. NB reports employment by Prelude Therapeutics, Inc. DMD reports current employment by Sanofi US and no other competing interests related to the work presented. ELC reports research funding or non-financial support from PICI, AstraZeneca, GuardantHealth, United Healthcare Group (UHG), Tempus, C2i, Oncocyte, Merck, ChipDX, Becton Dickinson and NIH/NCI. She received personal fees from BMS and Foundation Medicine. TBK reports grants from Bristol-Meyers Squibb, Eli Lilly Corporation, Genentech, Taiho, Xencor, Tempest Therapeutics, BioNTech, Boehringer Ingelheim, Replimune, Genfit, Totus Medicines, and PMV Pharma outside of the submitted work and personal fees from Incyte, AstraZeneca, Nucorion, Hepatitis B Foundation, Pfizer, Taiho, and Synnovation Therapeutics, all outside of the submitted work. SMB is an employee and shareholder of Personalis Inc, a company which performed genomic sequencing and joint research for the work in this publication. ROC is also an inventor on US patent number 09183496 issued to Personalis, which describes the genomic analyses in the Personalis sequencing platform used to sequence the samples in this study. He is also an employee of Personalis, a company which performed genomic sequencing and joint research for the work in this publication. LJP reports employment with Noetik, Inc. AHK reports research funding (paid directly to his institution) from Apexigen, Astellas, BMS, Celgene, Crystal Genomics, LEAP Therapeutics, Roche/Genentech, Verastem, the Pancreatic Cancer Action Network (PanCAN), and Parker Institute for Cancer Immunotherapy (PICI); and consulting/DSMB for Aadi, Arcus, Eisai, Fibrogen, Grail, Ipsen, Merus, and Roche/Genentech. JRE reports research funding (institutional) from Oncolys, Gilead, Amgen, Hutchmed, Merck, Arcus and consulting/advisory board fees from Merck, Oncolys, Exelixis, Daiichi Sanyko. Additionally, she reports employment and stock for her spouse at Janssen and Merus. PSW, DMM, SMP, CNS, ZW, MY, DB, WL, SC, CC, QL, AAA, URT, PJO, CJS, HS, RAW, JPL and CRC report no competing interests related to the work presented.
