## [Peer Review File · Nature Communications]

Circulating *KRAS* G12D but not G12V is associated with survival in metastatic pancreatic ductal adenocarcinomaThis manuscript has been previously reviewed at another journal that is not operating a transparent peer review scheme. This document only contains reviewer comments and rebuttal letters for versions considered at *Nature Communications*.

REVIEWER COMMENTS

Reviewer #2 (Remarks to the Author):

This manuscript presents an interesting study on the differential detection of various KRAS mutations in the plasma of patients with pancreatic cancer. The main critique of this paper is that this result could be due to a different detection sensitivity between digital PCR for G12V and G12D. The authors do not provide sufficient results in their manuscript to support this point. I would consider this work valid if, on at least one of the two sets, validation by another detection method showed the same result, such as using NGS as an example.

Reviewer #4 (Remarks to the Author):

In the study by Till et al., the authors demonstrated the specific association of circulating plasma KRAS G12D DNA levels with survival of patients with previously untreated first-line metastatic pancreatic ductal adenocarcinoma (mPDAC), which provides certain evidence of its potential application in the clinic.

Despite that, there are some critical concerns, as listed below:

1. The survival analysis results may be inaccurate. While running a Cox multivariate model to account for potential confounding factors is acceptable, it's essential to consider other critical factors that could influence survival. These factors might include the response to first or second-line treatments, treatment-associated side effects, tumor burden, and economic status. Hence, it is advisable to incorporate these potential factors into the analysis as well.
2. I propose pooling the PRINCE and SOC cohorts to investigate whether KRAS G12D (versus G12V) DNA levels continue to serve as an independent predictor of survival.
3. Could the authors provide potential biological explanations for why circulating plasma DNA levels of KRAS G12D, rather than G12V, are correlated with the survival of mPDAC patients? For instance, could KRAS G12D be linked to improved treatment response?

RESPONSE TO REVIEWERS' COMMENTS

Reviewer comments for Till, et al., “Baseline and early on-therapy circulating plasma KRAS G12D but not G12V DNA levels are associated with survival of patients with previously untreated metastatic pancreatic ductal adenocarcinoma,” are listed below with our replies in blue.

Reviewer #2

This manuscript presents an interesting study on the differential detection of various KRAS mutations in the plasma of patients with pancreatic cancer. The main critique of this paper is that this result could be due to a different detection sensitivity between digital PCR for G12V and G12D. The authors do not provide sufficient results in their manuscript to support this point. I would consider this work valid if, on at least one of the two sets, validation by another detection method showed the same result, such as using NGS as an example.

We agree about the importance of ruling out the possibility that our finding of a variant-specific difference in the association of plasma ctKRAS levels with survival is due to a difference in the sensitivity of our ddPCR assay to detect *KRAS* G12V vs G12D. As we described in the previous rebuttal letter, we have two sets of results to suggest that this is highly unlikely to be the case:

1. No difference in plasma/tissue concordance for *KRAS* G12D vs G12V: Given that matched tissue NGS results were available for the PRINCE cohort (no tissue NGS was performed for the SOC cohort), we analyzed plasma/tissue concordance, i.e., the percent of cases for which the same variant was detected in plasma by ddPCR and in matched tissue by NGS. If our study results were due to a difference in the sensitivity of the ddPCR assay to detect *KRAS* G12D vs G12V in plasma, then one would expect to see a difference in the plasma/tissue concordance for the two variants. For example, if our ddPCR G12V assay were less sensitive than the assay for G12D, one would expect a lower overall plasma/tissue concordance for G12V, with a higher rate of patients for whom G12V is detected in tissue but not in matched plasma. However, our analysis found no difference (Fisher’s exact test $p > 0.9999$) in G12D concordance for plasma vs matched tissue (89.3% or 25/28) compared to G12V concordance (87.5% or 21/24). Importantly, for the 3 discordant G12D’s and the 3 discordant G12V’s, all were found by our plasma ddPCR assay and not by NGS of matched tissue. Sentences describing these latter analyses were added to the Methods section. The raw data for this concordance analysis is included in the detailed table in **Supplemental Table 1**. These results are consistent with a lack of difference in the sensitivity of our ddPCR assay to detect *KRAS* G12V vs G12D.
2. Sensitivity analysis of ddPCR levels of detection (LODs): To provide further evidence that our results are unlikely to hinge on a variant-specific difference in our assay’s sensitivity, we re-analyzed our ddPCR results at three levels of detection (LODs). Given that our ddPCR assay has a lower LOD than most plasma NGS platforms, the aim of this analysis is to demonstrate that our results hold up at the higher LODs typically utilized by commercially available clinical plasma ctDNA testing such as that conducted by GuardantHealth [Lanman RB, et al., PLOS One 2015]. Shown below are results for the combined PRINCE and SOC cohorts at the LOD of 0.04% used throughout our current figures, as well as at 0.10% and 0.25%, typically used as LODs for plasma NGS results. The table immediately below shows hazard ratios and log-rank p-values for the survival association (OS and PFS) for baseline ctKRAS VAF by variant (G12D only and G12V only) and dichotomized at the median. These results have been added to the manuscript at the top of **Supplemental Table 5**:

Survival	LoD	G12D	G12V
OS	0.04%	HR 2.95 (95% CI 1.67-5.20) p=0.0001	HR 1.38 (95% CI 0.74-2.59) p=0.3083
	0.10%	HR 2.62 (95% CI 1.47-4.66) p=0.0007	HR 1.38 (95% CI 0.74-2.59) p=0.3083
	0.25%	HR 1.98 (95% CI 1.08-3.62) p=0.0244	HR 1.31 (95% CI 0.68-2.53) p=0.4203
PFS	0.04%	HR 2.91 (95% CI 1.63-5.17) p=0.0002	HR 0.82 (95% CI 0.45-1.53) p=0.5355
	0.10%	HR 2.86 (95% CI 1.58-5.20) p=0.0003	HR 0.82 (95% CI 0.45-1.53) p=0.5355
	0.25%	HR 1.80 (95% CI 0.97-3.34) p=0.058	HR 0.66 (95% CI 0.34-1.28) p=0.209

In addition (and as shown below), we performed the Cox analysis for baseline log ctKRAS VAF as a continuous variable in both univariate and multivariate analysis at the three LODs. These results were also added to **Supplemental Table 5**:

LoD = 0.04%	G12D N=53				G12V N=32				
	Univariate		Multivariate		Univariate		Multivariate		
Variable	HR [95% CI]	p-value	HR [95% CI]	p-value	HR [95% CI]	p-value	HR [95% CI]	p-value	
OS	logVAF	1.77 [1.27-2.48]	0.001	2.27 [1.44-3.60]	0.000	1.06 [0.66-1.69]	0.810	1.41 [0.79-2.53]	0.245
	Age	1.00 [0.97-1.03]	0.998	1.03 [0.99-1.07]	0.116	1.05 [1.00-1.09]	0.053	1.03 [0.97-1.09]	0.323
	SexOF1M	1.97 [1.08-3.57]	0.026	3.31 [1.53-7.17]	0.002	0.42 [0.20-0.92]	0.029	0.50 [0.20-1.24]	0.135
	ECOGPS	1.92 [1.23-2.97]	0.004	1.70 [1.05-2.74]	0.030	1.89 [1.00-3.54]	0.049	2.28 [0.98-5.30]	0.055
	logSOD [‡]	3.44 [1.07-11.04]	0.038	0.43 [0.09-2.13]	0.303	2.95 [0.23-37.35]	0.404	0.23 [0.01-5.30]	0.355
	logCA19-9	1.21 [0.97-1.52]	0.092	1.04 [0.84-1.29]	0.717	0.80 [0.57-1.13]	0.206	0.92 [0.61-1.39]	0.690
PFS	logVAF	1.65 [1.18-2.30]	0.003	2.05 [1.35-3.10]	0.001	0.96 [0.64-1.45]	0.854	1.25 [0.77-2.03]	0.357
	Age	1.00 [0.97-1.03]	0.920	1.02 [0.99-1.06]	0.253	1.05 [1.00-1.10]	0.069	1.03 [0.98-1.09]	0.274
	SexOF1M	2.30 [1.26-4.19]	0.006	3.27 [1.58-6.78]	0.001	0.59 [0.28-1.24]	0.165	1.10 [0.42-2.88]	0.847
	ECOGPS	1.40 [0.92-2.14]	0.120	1.25 [0.78-2.00]	0.346	2.17 [1.05-4.49]	0.037	1.98 [0.89-4.41]	0.096
	logSOD [‡]	2.50 [0.81-7.77]	0.112	0.45 [0.10-1.95]	0.285	2.42 [0.18-32.77]	0.506	1.02 [0.06-17.89]	0.992
	logCA19-9	1.15 [0.92-1.42]	0.218	1.00 [0.81-1.24]	0.972	0.70 [0.50-0.98]	0.038	0.66 [0.42-1.04]	0.071

LoD = 0.10%	G12D N=50				G12V N=32				
	Univariate		Multivariate		Univariate		Multivariate		
Variable	HR [95% CI]	p-value	HR [95% CI]	p-value	HR [95% CI]	p-value	HR [95% CI]	p-value	
OS	logVAF	1.71 [1.18-2.47]	0.004	2.22 [1.38-3.58]	0.001	1.06 [0.66-1.69]	0.810	1.41 [0.79-2.53]	0.245
	Age	1.00 [0.97-1.03]	0.939	1.03 [0.99-1.07]	0.135	1.05 [1.00-1.09]	0.053	1.03 [0.97-1.09]	0.323
	SexOF1M	1.99 [1.08-3.67]	0.028	3.31 [1.50-7.30]	0.003	0.42 [0.20-0.92]	0.029	0.50 [0.20-1.24]	0.135
	ECOGPS	1.77 [1.11-2.81]	0.016	1.70 [1.03-2.79]	0.037	1.89 [1.00-3.54]	0.049	2.28 [0.98-5.30]	0.055
	logSOD [‡]	2.67 [0.77-9.20]	0.121	0.48 [0.10-2.40]	0.375	2.95 [0.23-37.35]	0.404	0.23 [0.01-5.30]	0.355
	logCA19-9	1.15 [0.92-1.44]	0.231	1.03 [0.82-1.28]	0.825	0.80 [0.57-1.13]	0.206	0.92 [0.61-1.39]	0.690
PFS	logVAF	1.68 [1.16-2.42]	0.006	2.15 [1.38-3.34]	0.001	0.96 [0.64-1.45]	0.854	1.25 [0.77-2.03]	0.357
	Age	1.00 [0.97-1.03]	0.790	1.02 [0.98-1.05]	0.360	1.05 [1.00-1.10]	0.069	1.03 [0.98-1.09]	0.274
	SexOF1M	2.10 [1.14-3.87]	0.017	3.12 [1.49-6.57]	0.003	0.59 [0.28-1.24]	0.165	1.10 [0.42-2.88]	0.847
	ECOGPS	1.34 [0.86-2.09]	0.198	1.33 [0.81-2.17]	0.259	2.17 [1.05-4.49]	0.037	1.98 [0.89-4.41]	0.096
	logSOD [‡]	1.99 [0.58-6.82]	0.273	0.46 [0.10-2.07]	0.313	2.42 [0.18-32.77]	0.506	1.02 [0.06-17.89]	0.992
	logCA19-9	1.11 [0.89-1.38]	0.374	1.00 [0.81-1.25]	0.973	0.70 [0.50-0.98]	0.038	0.66 [0.42-1.04]	0.071

LoD = 0.25%	G12D N=43				G12V N=30				
	Univariate		Multivariate		Univariate		Multivariate		
Variable	HR [95% CI]	p-value	HR [95% CI]	p-value	HR [95% CI]	p-value	HR [95% CI]	p-value	
OS	logVAF	1.68 [1.02-2.79]	0.042	2.08 [1.18-3.66]	0.011	0.93 [0.53-1.64]	0.806	1.47 [0.71-3.03]	0.303
	Age	1.00 [0.97-1.03]	0.987	1.03 [0.99-1.07]	0.149	1.04 [0.99-1.09]	0.118	1.03 [0.97-1.09]	0.316
	SexOF1M	2.21 [1.14-4.29]	0.020	3.25 [1.40-7.51]	0.006	0.47 [0.22-1.04]	0.061	0.50 [0.20-1.27]	0.146
	ECOGPS	1.54 [0.82-2.89]	0.182	1.59 [0.84-3.01]	0.158	1.68 [0.88-3.20]	0.115	2.28 [0.93-5.58]	0.071
	logSOD [‡]	1.45 [0.35-6.00]	0.605	0.38 [0.07-2.07]	0.262	2.69 [0.23-31.97]	0.433	0.19 [0.01-5.19]	0.328
	logCA19-9	1.08 [0.86-1.36]	0.493	1.02 [0.81-1.28]	0.882	0.80 [0.57-1.12]	0.201	0.96 [0.63-1.45]	0.840
PFS	logVAF	1.76 [1.05-2.95]	0.031	1.98 [1.15-3.43]	0.014	0.81 [0.50-1.32]	0.390	0.95 [0.48-1.88]	0.888
	Age	1.00 [0.96-1.03]	0.805	1.02 [0.98-1.06]	0.298	1.03 [0.98-1.09]	0.214	1.01 [0.95-1.08]	0.689
	SexOF1M	2.45 [1.26-4.76]	0.008	3.21 [1.42-7.27]	0.005	0.70 [0.33-1.51]	0.366	1.45 [0.48-4.34]	0.511
	ECOGPS	1.32 [0.71-2.44]	0.380	1.42 [0.75-2.67]	0.283	1.78 [0.85-3.71]	0.126	1.48 [0.59-3.70]	0.399
	logSOD [‡]	1.58 [0.36-6.93]	0.543	0.48 [0.09-2.73]	0.410	2.12 [0.17-27.14]	0.564	1.53 [0.07-32.84]	0.787
	logCA19-9	1.06 [0.85-1.33]	0.597	0.99 [0.79-1.25]	0.957	0.68 [0.48-0.96]	0.027	0.62 [0.38-1.02]	0.060

[‡]SOD = Sum of diameters for all target lesions

Taken together, these sensitivity analyses strongly suggest that a variant-specific difference in the sensitivity of our ddPCR assay is not the driver of our results, nor would an NGS test be likely to yield conflicting results. We hope the information provided here convincingly demonstrates that these results are not merely artifacts of using a sensitive ddPCR assay and have added the following language to the last paragraph of the

Discussion: “While ours is the first study, to our knowledge, demonstrating variant-specific prognostic value of a commercially available ddPCR assay, it will be essential to validate this result using an orthogonal assay, such as widely used plasma targeted NGS testing.”

Per the editor’s specific direction to us in her email, we will not include further validation of either of our cohorts.

Reviewer #4

In the study by Till et al., the authors demonstrated the specific association of circulating plasma KRAS G12D DNA levels with survival of patients with previously untreated first-line metastatic pancreatic ductal adenocarcinoma (mPDAC), which provides certain evidence of its potential application in the clinic. Despite that, there are some critical concerns, as listed below:

1. The survival analysis results may be inaccurate. While running a Cox multivariate model to account for potential confounding factors is acceptable, it's essential to consider other critical factors that could influence survival. These factors might include the response to first or second-line treatments, treatment-associated side effects, tumor burden, and economic status. Hence, it is advisable to incorporate these potential factors into the analysis as well.

We appreciate the reviewer’s enthusiasm for the Cox multivariate modelling we performed and the interest in including additional variables. In fact, we had previously discussed this possibility with our biostatisticians but were advised that going beyond the variables we currently include in the multivariate analysis, beyond log ctKRAS VAF, is the maximum they were comfortable with given the size of our cohorts. As far as the specific variables listed by this reviewer:

- The “response to first or second-line treatments” is addressed by the inclusion of analysis for the surrogate endpoint of progression-free survival (PFS) in **Supplemental Table 4** as this is essentially a measure of response to first-line therapy.
- Adverse events were not rigorously or comprehensively collected during therapy for the SOC cohort. In the PRINCE cohort, >98% of patients experienced at least one treatment-related adverse event (TRAE) [as reported in O’Hara M et al., *Lancet Oncol* 2021] and, therefore, including the occurrence of a TRAE into the multivariate model is highly unlikely to impact the results.
- Tumor burden is already included in the multi-variate analysis for the PRINCE cohort for which the “logSOD” (log of the sum of diameters; the sum of diameters of RECIST target lesions) is included as an independent variable (**Supplemental Table 4, Supplemental Figure 2**). The VAF term in the multivariate analysis remains significant when including “logSOD” (which itself is not significant), demonstrating that tumor burden is not confounding the *KRAS* G12D VAF association with OS/PFS.
- Data about economic status was not collected for our cohorts.

Altogether, we propose that our multivariate model considers the maximum allowable number of variables commonly considered in such models for this tumor type and for variables without large amounts of missing data. This suggests that the model is indeed accurate to the extent possible.

2. I propose pooling the PRINCE and SOC cohorts to investigate whether *KRAS* G12D (versus G12V) DNA levels continue to serve as an independent predictor of survival.

We thank the reviewer for this suggestion and have added pooled analysis of the PRINCE and SOC cohorts for all relevant results. In all cases, this new analysis fully supports the conclusion that KRAS G12D (versus G12V) ctDNA levels serve as an independent predictor of survival. Shown below are each of the new pooled figures and where they have been added to the revised manuscript.

First, we demonstrate for the pooled PRINCE and SOC cohorts that there is no significant difference in OS (left) or PFS (right) when comparing patients whose tumors express G12D (green curve) or G12V (dark grey curve). This result is consistent with the analysis of PRINCE and SOC cohorts individually, and has been added to **Supplemental Figure 4** as panels C and F:

Next, we demonstrate that when OS and PFS are analyzed for the combined cohort with respect to ctKRAS VAF for all ctKRAS variants detected, VAF dichotomized at the median value is significant for both survival endpoints. This result is consistent with the analysis of PRINCE and SOC cohorts individually, and has been added to **Supplemental Figure 5** as panels C and F:

Next, we demonstrate no significant difference in ctKRAS VAF (P=0.8728) when comparing patients whose tumors express ctKRAS G12D (green dots) versus G12V (dark grey dots). This result is consistent with the analysis of PRINCE and SOC cohorts individually, and has been added as **Supplemental Figure 6C**:

Combined PRINCE and SOC cohorts

Next, we analyzed the combined PRINCE and SOC cohort to confirm that ctKRAS VAF for G12D remains significantly associated with OS (top panel A) and PFS (bottom panel C), but these associations are not significant for ctKRAS G12V (top panel B and bottom panel D, respectively). This result is consistent with the analysis of PRINCE and SOC cohorts individually, and has been added as new **Supplemental Figure 7**:

Combined PRINCE and SOC cohorts

Next, we demonstrate that the above findings remain true when PRINCE and SOC are combined in a Cox analysis for baseline log ctKRAS VAF as a continuous variable. ctKRAS G12D remains significantly associated with OS and PFS, but G12V does not, when analyzed as a continuous variable either alone or in a multivariate analysis. These results are consistent with the analysis of PRINCE and SOC cohorts individually, and has been added to the bottom of **Supplemental Table 4**:

		Combined PRINCE and SOC Patients							
		G12D (n=53)				G12V (n=32)			
		Univariate		Multivariate		Univariate		Multivariate	
		HR [95% CI]	p-value	HR [95% CI]	p-value	HR [95% CI]	p-value	HR [95% CI]	p-value
OS	logVAF	1.77 [1.27-2.48]	0.001	2.27 [1.44-3.60]	<0.001	1.06 [0.66-1.69]	0.810	1.41 [0.79-2.53]	0.245
	Age	1.00 [0.97-1.03]	0.998	1.03 [0.99-1.07]	0.116	1.05 [1.00-1.09]	0.053	1.03 [0.97-1.09]	0.323
	Sex	1.97 [1.08-3.57]	0.026	3.31 [1.53-7.17]	0.002	0.42 [0.20-0.92]	0.029	0.50 [0.20-1.24]	0.135
	ECOG PS	1.92 [1.23-2.97]	0.004	1.70 [1.05-2.74]	0.030	1.89 [1.00-3.54]	0.049	2.28 [0.98-5.30]	0.055
	logSOD ¹	3.44 [1.07-11.04]	0.038	0.43 [0.09-2.13]	0.303	2.95 [0.23-37.35]	0.404	0.23 [0.01-5.30]	0.355
	logCA19-9	1.21 [0.97-1.52]	0.092	1.04 [0.84-1.29]	0.717	0.80 [0.57-1.13]	0.206	0.92 [0.61-1.39]	0.690
PFS	logVAF	1.65 [1.18-2.30]	0.003	2.05 [1.35-3.10]	0.001	0.96 [0.64-1.45]	0.854	1.25 [0.77-2.03]	0.357
	Age	1.00 [0.97-1.03]	0.920	1.02 [0.99-1.06]	0.253	1.05 [1.00-1.10]	0.069	1.03 [0.98-1.09]	0.274
	Sex	2.30 [1.26-4.19]	0.006	3.27 [1.58-6.78]	0.001	0.59 [0.28-1.24]	0.165	1.10 [0.42-2.88]	0.847
	ECOG PS	1.40 [0.92-2.14]	0.120	1.25 [0.78-2.00]	0.346	2.17 [1.05-4.49]	0.037	1.98 [0.89-4.41]	0.096
	logSOD ¹	2.50 [0.81-7.77]	0.112	0.45 [0.10-1.95]	0.285	2.42 [0.18-32.77]	0.506	1.02 [0.06-17.89]	0.992
	logCA19-9	1.15 [0.92-1.42]	0.218	1.00 [0.81-1.24]	0.972	0.70 [0.50-0.98]	0.038	0.66 [0.42-1.04]	0.071

¹SOD = Sum of diameters for all target lesions

Next, we tested the possibility of a non-linear relationship between ctKRAS and survival by plotting the estimated restricted cubic spline function relating to the univariate log ctKRAS Cox models for the combined PRINCE and SOC cohort. These results are consistent with the analysis of PRINCE and SOC cohorts individually, and have been added to the bottom of **Supplemental Figure 8**.

Additionally, we included the combined PRINCE and SOC cohort in the sensitivity analysis of ddPCR levels of detection (LODs) (described above in the response to reviewer 2) and added as new **Supplemental Table 5**.

Finally, we demonstrate similar results for on-therapy ctKRAS clearance for the combined cohort below. This result is consistent with the analysis of PRINCE and SOC cohorts individually, and has been added as new **Supplemental Figures 10 and 11**:

All Combined PRINCE and SOC Patients

Combined PRINCE and SOC cohorts

Taken together, the above results for the combined PRINCE and SOC cohorts uphold our findings for each cohort individually. We acknowledge these findings with the addition of this sentence to the last paragraph of the Discussion: “Moreover, even when the two cohorts were combined, ctKRAS G12D, but not G12V, remained an independent predictor of survival.”

3. Could the authors provide potential biological explanations for why circulating plasma DNA levels of KRAS G12D, rather than G12V, are correlated with the survival of mPDAC patients? For instance, could KRAS G12D be linked to improved treatment response?

We agree that delving into the biological explanations for our results is an important next step. As a hypothesis, we could speculate that there are driver variant-specific differences in rates of necrosis and apoptosis that are associated with different levels of cfDNA shedding and thus alter the associations with survival. We do already show, using PFS as a surrogate for treatment response, that KRAS G12D is not uniquely linked to improved treatment response (**Supplementary Figure 4** reproduced below):

Progression-Free Survival

D All PRINCE patients

E All SOC patients

F Combined PRINCE and SOC patients

Overall, such questions are beyond the scope of our study and we agree with the editor's previous conclusion that additional experiments to this end were not necessary.

REVIEWERS' COMMENTS

Reviewer #4 (Remarks to the Author):

Congratulations, the authors have successfully addressed all my concerns.

Reviewer #5 (Remarks to the Author):

The authors have satisfied my concerns.